# Exosomal Long Non-Coding Ribonucleic Acid Ribonuclease Component of Mitochondrial Ribonucleic Acid Processing Endoribonuclease Is Defined as a Potential Non-Invasive Diagnostic Biomarker for Bladder Cancer and Facilitates Tumorigenesis via the miR-206/G6PD Axis

**DOI:** 10.3390/cancers15215305

**Published:** 2023-11-06

**Authors:** Yuting Gao, Xuan Wang, Huarong Luo, Chen Chen, Jing Li, Ruixin Sun, Dong Li, Zujun Sun

**Affiliations:** 1Department of Laboratory Medicine, Shanghai Tongji Hospital, Tongji University School of Medicine, Shanghai 200065, China; 2211531@tongji.edu.cn (Y.G.); olivia_chan@tongji.edu.cn (C.C.); qingshuiqiuqian@126.com (R.S.); 2Department of Pharmacy, Putuo People’s Hospital, School of Medicine, Tongji University, Shanghai 200060, China; wxandld186@163.com; 3Department of Urology, Shanghai Tongji Hospital, Tongji University School of Medicine, Shanghai 200065, China; 03824@tongji.edu.cn

**Keywords:** bladder cancer, RMRP, diagnostic biomarker, tumor progression, miR-206, G6PD

## Abstract

**Simple Summary:**

Bladder cancer (BLCA) is one of the most prevalent urinary cancers and novel non-invasive BLCA tumor biomarkers with high sensitivity and specificity are needed to support early diagnosis. Exosomal long non-coding RNAs (lncRNAs) are promising diagnostic biomarkers for BLCA. Our study suggests that the combined evaluation of lncRNA RMRP levels in urinary and plasma exosomes is an excellent potential non-invasive diagnostic biomarker for BLCA patients. Additionally, targeting the RMRP/miR-206/G6PD axis is a promising strategy for BLCA treatment.

**Abstract:**

Bladder cancer (BLCA) is one of the cancers that is highly sensitive to specific non-invasive tumor biomarkers that facilitate early diagnosis. Exosome-derived long non-coding RNAs (lncRNAs) hold promise as diagnostic biomarkers for BLCA. In this study, we employed RNA-sequencing to compare the expression patterns of lncRNAs in urine exosomes from three BLCA patients and three healthy individuals. RMRP displayed the most significant differential expression. Elevated RMRP expression levels were observed in urinary and plasma exosomes from BLCA patients compared with those from healthy individuals. RMRP exhibited significant associations with certain BLCA patient clinicopathological features, including tumor stage, poor prognosis, and tumor grade. Combined diagnosis using RMRP in urine and plasma exosomes demonstrated a superior diagnostic performance with receiver operating characteristic curve analysis. RMRP was found to be related to BLCA tumor progression and the cell migration and invasion processes via the miR-206/G6PD axis both in vitro and in vivo. Mechanistically, RMRP serves as an miR-206 sponge, as suggested by dual-luciferase reporter assays and RNA immunoprecipitation. Our study suggests that the combined diagnosis of RMRP in urinary and plasma exosomes can serve as an excellent non-invasive diagnostic biomarker for BLCA patients. Additionally, targeting the RMRP/miR-206/G6PD axis holds promise as a therapeutic strategy for BLCA.

## 1. Introduction

Bladder cancer (BLCA) is a highly invasive malignant tumor that affects the urinary system. Data from 2023 indicate that BLCA has become one of the top ten most prevalent cancers globally [1]. Currently, the clinical use of biomarkers, such as nuclear matrix protein 22 (NMP-22) and bladder tumor antigen (BTA), suffers from low sensitivity and the potential for false positives [2,3,4]. While cystoscopy remains the gold standard for diagnosing BLCA, it is an invasive procedure. Additionally, BLCA is often asymptomatic in its early stages, making cystoscopy an unsuitable early screening tool [5,6]. Patients in the early stages of BLCA typically do not exhibit noticeable symptoms, emphasizing the critical importance of early BLCA diagnosis for good prognosis [7,8]. Therefore, the identification of promising early molecular biomarkers holds significant potential for a better understanding of BLCA pathogenesis and enhancing its clinical treatment.

Exosomes are nanometer-sized vesicles (30–150 nm) that can be isolated from liquid biopsy samples. Their cargo can include DNA, microRNAs (miRNAs), long non-coding RNAs (lncRNAs), circular RNAs, lipids, and several types of proteins [9]. Exosomes are suitable diagnostic and prognostic biomarkers because they are not easily degraded by RNases [10]. LncRNAs are a class of RNA molecules greater than 200 nucleotides in length that do not encode protein. They have been shown to play key roles in the biological and pathological processes of tumors [11]. Several studies have demonstrated that exosomal lncRNAs in urine and blood can be used as diagnostic biomarkers in BLCA patients [12,13,14,15,16]. There is also a lack of non-invasive tumor biomarkers for diagnosing BLCA with high sensitivity in clinical practice [17]. It is therefore urgent to identify non-invasive biomarkers to diagnose BLCA at an earlier stage, which can allow for more timely treatment. Exosome testing is a novel liquid biopsy technique. Many studies have reported the usefulness of exosome testing in the early detection and prognostic evaluation of different types of malignancies [18,19,20]. Exosomes are involved in BLCA progression by encapsulating lncRNAs to mediate extracellular communication and can be used as potential biomarkers for BLCA detection. For example, lncRNA PTENP1, a group of three lncRNAs (MALAT1, PCAT-1, and SPRY4-IT1) [21,22], and a panel of three lncRNAs (PCAT-1, UBC1, and SNHG16) were encapsulated in BLCA urine or blood exosomes and could serve as essential biomarkers for disease diagnosis or prognosis [13].

Recently, researchers have demonstrated that lncRNAs are frequently dysregulated in various malignant tumors [23]. LncRNAs reportedly contribute to cancer development by regulating several cellular processes that are crucial to tumorigenesis, such as cell growth, invasion, migration, and apoptosis [24,25,26]. Several studies have shown that certain lncRNAs can serve a competitive endogenous RNA (ceRNA) role by regulating the biological function or expression of specific miRNAs. For example, lncRNA LINC00963 can promote tumorigenesis and radioresistance by acting as a ceRNA of miR-124-3p and miR-204-3p [27,28]. However, the clinical significance and biological mechanism of most lncRNAs in BLCA remain unknown.

In a previous study, we examined the lncRNAs that are upregulated in urine exosomes using RNA-sequencing (RNA-seq), finding that the RNA component of mitochondrial RNA processing endoribonuclease (RMRP) was the most highly expressed in the BLCA patient group relative to the normal controls [15]. RMRP is a lncRNA that plays an essential role in the development of tumors [29,30,31]. For example, RMRP has been reported to play a pro-cancer role in non-small cell lung cancer and colorectal cancer [32,33]. It has a regulatory role in RNA processing in both ribosomal compartments and mitochondria [34]. In this study, we aimed to investigate if RMRP can be used as a potential biomarker for BLCA diagnosis and to study whether RMRP is specifically involved in BLCA progression.

## 2. Materials and Methods

### 2.1. Clinical Studies

All plasma, urine, and tissue samples were collected from patients who were diagnosed with BLCA using histopathological results between October 2019 and February 2022 at Shanghai Tongji Hospital and Fudan University Shanghai Cancer Center. The clinical information of the patients is provided in Appendix A. Additionally, individuals who received healthy physical examinations were recruited from the health check-up center of Shanghai Tongji Hospital and used as a control group. The Ethics Committee of Shanghai Tongji Hospital (No.2021-KYSB-064) and Fudan University Shanghai Cancer Center (No. 050432-4-1911D) approved this study.

### 2.2. Exosome Isolation

Exosomes were isolated from urine and plasma samples using a protocol adapted from the exosome extraction kit. Exosome isolation from urine was performed as described previously [15]. The ExoQuick kit (BB-39012, Bestbio, Shanghai, China) was used to extract exosomes from all plasma and urine samples. To confirm successful extraction, a combination of transmission electron microscopy (TEM), nanoparticle tracking analysis (NTA), and Western blot (WB) analysis was used to identify exosomes (please see the following sections for detailed procedures). The TEM results demonstrated that urine exosomes and plasma exosomes had cup-like depressions and an intact envelope (Figure 1A), while the NTA results showed urine exosomes and plasma exosomes ranging in size from 30 to 150 nm (Figure 1B). WB analysis was used to confirm the expression of exosomal protein markers. The results showed that the exosome-positive markers CD9 and TSG101 were expressed in both urinary and plasma exosomes, whereas the exosome-negative marker MG130 was not expressed in exosomes but only in cytosolic proteins (Figure 1C). These data confirmed the successfully isolation of exosomes from the samples. Plasma samples were centrifuged at 3000× *g* for 15 min at 4 °C to remove cells and debris, then centrifuged at 10,000× *g* for 20 min at 4 °C to isolate large microvesicles. Next, the supernatant was mixed with Solution A and Solution B from the exosome extraction kit, then stored overnight at 4 °C. Finally, the precipitate obtained after centrifugation at 10,000× *g* for 1 h at 4 °C (the exosomes) was resuspended in 50 μL of Solution C.

### 2.3. TEM

For TEM analysis, a 20 µL suspension of exosomes was placed on a 400-mesh carbon-coated copper grid, then negatively stained with a 2% phosphotungstic acid solution for 3 min. The copper grids were cleaned in double-distilled water and allowed to dry at room temperature. Subsequently, samples were observed and images were captured using TEM (Thermo Fisher Scientific, Waltham, MA, USA).

### 2.4. NTA

The NanoSight LM10 system (Malvern Instruments Ltd., Malvern, UK) was used to detect the concentration and size of the isolated exosomes.

### 2.5. WB Analysis

Total protein was extracted from the samples with radio immunoprecipitation assay (RIPA) lysis buffer (Thermo Fisher Scientific). Protein concentrations were determined using a bicinchoninic acid (BCA) protein assay kit (Thermo Fisher Scientific). Equal amounts of exosome protein samples were separated by 10% SDS-PAGE, which were then transferred onto polyvinylidene fluoride (PVDF) membranes (Millipore, Billerica, MA, USA). The membranes were blocked with 5% non-fat milk in TBST buffer and incubated with primary antibodies against CD9 (1:1000, 13174S; Cell Signaling Technology (CST), Danvers, MA, USA), TSG101 (1:1000, Ab83; Abcam, Cambridge, UK), MG130 (1:1000, 12480S; CSTUSA); G6PD (1:1000, 12263S; CST, USA) and β-Actin (1:1000, 4967S; CST, USA) overnight at 4 °C. The membranes were then incubated with a secondary antibody (1:2000, Santa Cruz Biotechnology, Santa Cruz, CA, USA) for 1 h at room temperature. Finally, the blots were visualized with Immobilon™ Western Chemiluminescent HRP Substrate (Millipore, USA) and related data were analyzed by the Image LabTM 3.0 software (Image Lab Software; Bio-Rad Laboratories, Inc., Hercules, CA, USA).

### 2.6. RNA-seq Analysis

RNA was extracted from urinary exosomes from three BLCA patients and three healthy controls. RNA-seq was performed using an Illumina Novaseq 6000 system (San Diego, CA, USA) by Shanghai Kangcheng Biotechnology Co., Ltd. (Shanghai, China). Differences in gene expression levels were analyzed to screen for differentially expressed lncRNAs between BLCA patients and healthy controls. LncRNAs with an adjusted *p*-value < 0.05 and fold change >1.5 were considered differentially expressed lncRNAs.

### 2.7. Quantitative Reverse Transcription Polymerase Chain Reaction (RT-qPCR)

Total (exosomal/cellular/tissue) RNA was extracted using TRIzol Reagent (Invitrogen, Carlsbad, CA, USA) according to the manufacturer’s instructions. Free RMRP was extracted from plasma and urine using RNA fast 2000 (Shanghai Feijie, Shanghai, China). The concentration and purity of RNA samples were measured using a Nanodrop 2000 (Thermo Fisher Scientific, USA). The PrimeScript™ RT kit and gDNA Eraser (Takara, Kusatsu, Japan) were used for the reverse transcription of RNA to cDNA. The RT-qPCR was performed using qPCR SYBR Green mix (Bio-Rad, Hercules, CA, USA) with an Applied Biosystems 7300 real-time PCR system (Thermo Fisher Scientific,). The 2^−ΔΔCt^ method was employed to calculate the relative RNA expression levels. Primer kits for miRNA analysis was obtained from RiboBio (Guangzhou, China). The primer sequences for RMRP, 18S, and GAPDH can be found in Appendix A.

### 2.8. Cell Culture

Human bladder cancer cell lines (BIU-87 and 5637), and the HEK-293T cell line, were obtained from the American Type Culture Collection (ATCC, Manassas, VA, USA). BIU-87 and 5637 cells were cultured in RPMI-1640 medium (Invitrogen, USA), while HEK-293T cells were cultured in DMEM (Gibco, Waltham, MA, USA). All cell lines were supplemented with 10% fetal bovine serum (FBS; Gibco, USA) and maintained in an atmosphere containing 5% CO_2_.

### 2.9. Cell Transfection

The cells were plated in 6-well plates and transfected once they reached 50%–60% confluence. Lentivirus encoding the lncRNA RMRP was obtained from Shanghai Jikai Gene Chemical Technology Co, Ltd. (Shanghai, China). Transfection was performed according to the manufacturer’s instructions using the appropriate reagents. Lipofectamine™ 3000 reagent (Invitrogen, USA) was utilized as the transfection reagent for the plasmids.

### 2.10. Cell Counting Kit-8 (CCK8) Assay

Cells were seeded at a density of 2000 cells per well in a 96-well plate (Corning, Corning, NY, USA). At the specified time points (24 h, 48 h, 72 h), 10 μL of CCK-8 reagent (Dojindo, Kumamoto, Japan) was added to the culture medium in each well. The cells were then incubated at 37 °C for 2 h. The optical density (OD) was measured at 450 nm using a microplate reader (Synergy H4 Hybrid Reader, BioTek, Winooski, VT, USA). These experiments were repeated three times.

### 2.11. Transwell Assays

Cell migration and invasion abilities were assessed using Transwell assays. BIU-87 and 5637 cells were suspended in serum-free medium, and 1 × 10^5^ cells were seeded into the upper chambers of Transwell inserts (Corning). A culture medium containing 10% FBS was added to the lower chamber to serve as a chemoattractant for facilitating cell migration assays. Following an incubation period of 24–48 h, the Transwell chambers were meticulously immobilized using methanol, followed by staining for 15 min with 5% crystal violet (Kaigen, Linyi, China). After rinsing three times with phosphate-buffered saline (PBS), the cells that had successfully migrated were meticulously captured through photographic documentation and quantified across five distinct fields. These experiments were repeated three times. For the cell invasion assay, the same procedure was conducted, but the Transwell inserts were pre-coated with a Matrigel mixture before cell seeding.

### 2.12. Colony Formation Assay

BIU-87 and 5637 cells were seeded in six-well plates at a density of 2000 cells per well and cultured for 14 days. Subsequently, the cells were rinsed three times with PBS, fixed with methanol, and stained with 5% crystal violet for 15 min. Finally, the colonies were counted and recorded. These experiments were repeated three times.

### 2.13. 5-Ethynyl-20-deoxyuridine (EdU) Incorporation Assay

BIU-87 and 5637 cells were seeded into 24-well plates at a density of 5 × 10^4^ cells/well. Cell proliferation was assessed using the EdU incorporation assay kit (RiboBio, China), following the manufacturer’s protocol. The images were acquired using the Zeiss LSM880 NLO confocal microscope (Leica, Wetzlar, Germany).

### 2.14. Luciferase Reporter Assay

The wild-type (WT) or mutant versions of lncRNA RMRP and the 3′ untranslated region (UTR) of G6PD were synthesized and inserted into the luciferase reporter vector pmirGLO (Ribobio). These constructs were named RMRP-WT, RMRP-MUT, G6PD-3′UTR-WT, and G6PD-3′UTR-MUT, respectively. Mimics 206, mimics NC, inhibitor 206, and inhibitor NC were obtained from Guangzhou RiboBio Co., Ltd. HEK-293T cells were co-transfected with these plasmids along with miR-206 mimic or inhibitor. The relative luciferase activity was assessed using the Dual-Luciferase Assay Kit (Promega, Madison, WI, USA) following the manufacturer’s protocol. After a 48 h transfection period, luciferase activity was measured using the Dual-Luciferase Reporter Assay System (Promega).

### 2.15. RNA Immunoprecipitation (RIP) Assay

The Magna RIP™ RNA-binding protein immunoprecipitation kit (Millipore) was employed to investigate the interaction between lncRNA RMRP and miR-206. HEK-293T cells were lysed using a complete RIP buffer. The RIP assay used antibodies such as anti-AGO2 and control IgG (Millipore, USA). AGO2 is an essential component of the RNA-induced silencing complex (RISC). The RNA present in immunoprecipitate tests was used for cDNA synthesis, followed by evaluation using RT-qPCR.

### 2.16. Detection of Exosome-RMRP Stability

To determine the stability of exosomal RMRP in urine and plasma, one urine and plasma sample were randomly selected, each divided into 11 aliquots, and repeatedly frozen and thawed between −80 °C and room temperature (0 cycle, 2 cycles, 4 cycles, and 8 cycles) or had an extended incubation at room temperature for 6, 12, 18, or 24 h. In addition, the remaining three urine and plasma samples were incubated with RNase A purchased from Sigma (Cambridge, MA, USA) at 1 μg/mL alone or in combination with 0.1% Triton ×100 (Beyotime, Shanghai, China) for 30 min at room temperature.

### 2.17. Enzyme-Linked Immunosorbent Assay (ELISA) and Colloidal Gold Immunochromatography (GICA)

BTA levels in plasma and NMP-22 levels in urine were detected using the recommended procedures included in the respective ELISA kit instructions (Cloud-Clone CORP., Wuhan, China). Additionally, urine exosome-NMP-22 was detected using the Alere NMP22^®^ BladderChek^®^ Test based on GICA.

### 2.18. Urine Cytology Determination

Sediment was obtained by centrifugation of mid-segment urine at 1500 rpm for 15 min at room temperature. Three cytopathologists performed the examination, with the sample being classified as positive only if all three experts found BLCA cells. All other cases were classified as negative [35].

### 2.19. In Vivo Animal Experiments

Four-to-six-week-old male nude mice were purchased from Shanghai Sippr BK Laboratory Animal Co. Ltd. (Shanghai, China). All animal experiments were performed according to protocols approved by Shanghai Tongji Hospital Experimental Animal Care Commission. BIU-87 tumor cells were injected on the right flank of nude mice. In all experiments, tumor growth was measured by caliper once a week. Tumor volumes were calculated using the following formula: tumor volume = (length × width^2^)/2. Lack of survival was defined as death or tumor size > 2000 mm^3^. After 30 days, the tumors were excised, paraffin-embedded, and subjected to hematoxylin and eosin (H and E) staining using the Solarbio HE Staining Kit (G1120, Beijing, China).

### 2.20. Immunohistochemistry (IHC) Staining

Formalin-fixed paraffin-embedded (FFPE) tumor tissues were subjected to IHC staining. Firstly, the sections were deparaffinized and rehydrated, followed by exposure to 3% H_2_O_2_ in methanol to block endogenous peroxidase activity. Subsequently, the sections were blocked with 1% BSA for 30 min at room temperature. After blocking, the sections were incubated overnight at 4 °C with primary antibodies, including Ki67 (1:1000, 9449S; CST) and PCNA (1:1000, 13110S; CST, USA). Following primary antibody incubation, peroxidase-conjugated secondary antibodies (ChemMate DAKO EnVision Detection Kit (Glostrup, Denmark), Peroxidase/DAB, Rabbit/Mouse, USA) and detection reagents were applied to the sections.

### 2.21. Statistical Analysis

Statistical analysis was performed using SPSS (IBM SPSS Statistics, version 25.0, SPSS IBM, Armonk, NY, USA). The Kolmogorov–Smirnov test was used to analyze the data distribution of each sample group. The *t*-test or Wilcoxon rank sum test was used to compare the two groups. Numeration data were analyzed using the χ^2^ test. Pearson’s correlation was used to examine the correlation. Receiver operating characteristic (ROC) curves were established to evaluate the diagnostic value. The “pROC” package in R version 3.6.3 was used to calculate the area under the curve (AUC), sensitivity, specificity, cutoff value, youden index, positive predictive value (PPV), and negative predictive value (NPV). All statistical analyses were performed using GraphPad Prism software (GraphPad Prism version 8.3.1 for Windows; GraphPad Software, www.graphpad.com (accessed on 3 January 2022)). The Cancer Genome Atlas (TCGA) database was used to analyze the differential expression of RMRP. All data are presented as the mean ± standard error of the mean (SEM). ns: not significant, *p* ≥ 0.05, * *p* < 0.05, ** *p* < 0.01, and *** *p* < 0.001 were considered statistically significant.

## 3. Results

### 3.1. Urinary Exosome RMRP Was Highly Expressed in BLCA Patient Samples and Associated with a Poor Prognosis

First, RNA-seq analysis was performed to examine the potential differential expression of BLCA-specific genes. The cluster heat map in Figure 1D shows the significantly dysregulated lncRNAs in urinary exosomes from healthy controls and BLCA patients. The data revealed 158 differentially expressed lncRNAs between the urinary exosomes of BLCA patients and healthy controls, of which 145 were upregulated and 13 were downregulated in the BLCA samples (Figure 1E). Among the upregulated lncRNAs, RMRP was the most differentially expressed lncRNA (Figure 1E).

TCGA datasets showed that RMRP expression levels were upregulated in BLCA samples (Figure 1F). Furthermore, qRT-PCR assays suggested that RMRP expression levels were significantly increased in BLCA tumors compared with adjacent non-cancerous tissues (Figure 1G, Appendix A). While no significant difference in RMRP expression was observed between the blood and urine samples (as shown in Figure 1H), it is important to note that RMRP expression in urine and plasma exosomes from BLCA patients showed a remarkable increase compared with the levels observed in normal subjects (as shown in Figure 1H). Additionally, as the number of sample increased, the expression patterns of exosomal RMRP in both urine and plasma were found to be significantly higher in the BLCA group compared with both the normal and benign lesion groups (as shown in Figure 1I and Appendix A). These differences were statistically significant (*p* < 0.001), suggesting that RMRP expression in urinary and plasma exosomes may have diagnostic value for BLCA.

### 3.2. Diagnostic Efficacy of RMRP in Urinary and Plasma Exosomes in BLCA Patients

The BLCA patients’ clinical characteristics, including gender, age, tumor size, tumor number, tumor stage (T), lymph node metastasis, and tumor grade, are summarized in Table 1. Pearson’s χ^2^ test was performed to explore the associations between RMRP expression levels in urine and plasma exosomes and BLCA patient clinicopathological characteristics (Figure 2A,B, Appendix A). Then, BLCA patients were divided into low (*n* = 49) and high (*n* = 50) RMRP expression groups, using the median value of plasma or urine exosomal RMRP expression as the cut-off value (Figure 2A,B and Appendix A). These data indicated that plasma and urine exosomal RMRP expression was associated with tumor stage (T) (plasma: *p* = 0.0003, urine: *p* < 0.001) (Figure 2A,B). However, most other clinical characteristics had no significant relationships with RMRP expression levels in plasma and urine exosome (Appendix A). RMRP expression in urine was positively correlated with that in plasma exosomes (Figure 2C), but there was no significant correlation between RMRP expression levels in tumor tissues and those in plasma and urine exosome, which is possibly because of the limited number of samples (Appendix A).

We next explored the diagnostic efficacy of RMRP expression in urinary and plasma exosomes for distinguishing BLCA patients from healthy controls (Table 2). We first explored the diagnostic efficacy of NMP-22 (ELISA) (AUC = 0.688, 95% confidence interval (CI): 0.611–0.765, BTA (ELISA) (AUC = 0.626, 95% CI: 0.544–0.707), NMP-22 (GICA) (AUC = 0.695, 95% CI: 0.644–0.747) (Figure 2D), and urine cytology (AUC = 0.701, 95% CI: 0.651–0.751, Figure 2E). Similarly, ROC curve analysis was performed using 16.161 as the cut-off value and AUC values of RMRP in plasma and urine exosome were calculated. Compared with the expression of the abovementioned biomarkers, plasma and urine exosomal RMRP had higher AUC values (plasma: 0.870, 95% CI: 0.820–0.919; urinary: 0.720, 95% CI: 0.646–0.794) (Figure 2F). Overall, combined RMRP in urinary and plasma exosomes presented the best diagnostic efficacy (AUC = 0.889, 95% CI: 0.842–0.936) (Figure 2F).

Because good stability is a critical prerequisite for tumor biomarkers [35], we next verified the stability of RMRP in plasma and urinary exosomes. Here, RNase A was used to degrade RMRP and Triton × 100 was added to destruct the exosomal membrane [21]. RMRP expression in plasma and urinary exosomes was not affected after treatment with RNase A alone (Figure 2G). However, RMRP expression was significantly decreased following treatment with both RNase A and Triton ×100 (Figure 2G). These data indicated that the intact exosomes could protect RMRP from degradation and support its stability. Additionally, no changes in RMRP expression were detected after repeated freeze–thaw cycles (Figure 2H). Moreover, RMRP expression was also stable when plasma and urine samples were incubated at room temperature for various time periods (Figure 2I). In summary, RMRP in plasma and urine exosomes was stable and is a potential non-invasive diagnostic biomarker for BLCA.

### 3.3. RMRP Promotes BLCA Cells Proliferation, Migration, and Invasion

Given the high expression patterns of RMRP in BLCA tumor cells and tissues (Figure 3A), we next explored its biological function in BLCA cells. Short hairpin RNAs (shRNAs) were designed to target RMRP (sh-RMRP) and the knockdown efficiency was confirmed by RT-qPCR (Figure 3B). Then, sh-RMRP-1 and sh-RMRP-2 were used to construct lentiviral vectors to interfere with RMRP expression and used to generate stable cell lines that were screened with puromycin. CCK-8 assays demonstrated that RMRP knockdown could suppress cell proliferation in both 5637 and BIU-87 cells (Figure 3C). Limited cell cloning capabilities were also observed in BIU-87 and 5637 cells with RMRP knockdown (Figure 3D). Transwell assay results revealed that the migration and invasion abilities of BIU-87 and 5637 cells were weakened after RMRP knockdown (Figure 3E,F). Moreover, knockdown of RMRP significantly decreased the percentages of EdU-positive cells (Figure 3G,H). We also tested the effects of manipulating RMRP expression in a BLCA tumor-bearing model in vivo. BLCA tumor models of nude mice were established with BIU-87 cells transfected with sh-NC control and sh-RMRP (*n* = 5) (Figure 4A). Tumors of the sh-RMRP group grew more slowly than those of the sh-NC group (Figure 4B), according to the volumes and weights of the residual tumors (Figure 4C,D). Moreover, IHC assays were used to examine PCNA and Ki67 protein expression levels, both of which were lower in the sh-RMRP group compared with the sh-NC group (Figure 4E). Overall, these data suggested that RMRP could promote BLCA cell proliferation, migration, and invasion both in vitro and in vivo.

### 3.4. Positive Correlation between RMRP and G6PD Expression Levels in BLCA Cells

As reported previously, lncRNAs can interact with DNA, mRNAs, miRNAs, transcription factors, and a variety of functional proteins to regulate gene expression. They ultimately participate in various biological processes, including the cell cycle, apoptosis, and differentiation [36]. We next explored the RMRP target genes that were elevated in BLCA and associated with poor patient prognosis. Here, glucose-6-phosphate dehydrogenase (G6PD) was found to be highly expressed in BLCA tissues compared with normal tissues according to TCGA database (Figure 4F) and our clinical studies (Figure 4G). G6PD is the rate-limiting enzyme of the pentose phosphate pathway. Recently, multiple studies have demonstrated that elevated G6PD levels can promote cancer progression in numerous tumor types, including melanoma, leukemia, and BLCA [37,38]. Moreover, we observed a positive correlation between RMRP and G6PD expression levels in our clinical BLCA tissue samples (Figure 4H). Hence, we hypothesized that RMRP may target G6PD. In addition, the G6PD mRNA and protein expression levels were altered following RMRP knockdown in BIU-87 and 5637 cells (Figure 4I,J).

### 3.5. RMRP Serves as an miRNA Sponge of miR-206 to Target G6PD

LncRNAs can play a role of an miRNA sponge during tumor development [39,40]. RMRP can reportedly act as an miRNA sponge and thereby regulate the downstream target genes [41,42,43]. To further elucidate the mechanisms of RMRP in BLCA, we explored the ceRNA network of RMAP based on miRDB, miRTarBase, and TargetScan [44,45] (Figure 5A). From our findings, we proposed that RMRP can possibly regulate G6PD through miR-206 (Figure 5A). Transfection of cells with sh-RMRP markedly increased miR-206 expression levels (Figure 5B). RIP experiments with an anti-AGO2 antibody resulted in pulldown of both endogenous RMRP and miR-206, validating the indirect binding potential of these molecules (Figure 5C,D). Then, we used the online bioinformatics tool StarBase 2.0 (http://starbase.sysu.edu.cn/ (accessed on 3 January 2022)) to predict the binding sites of RMRP and miR-206 (Figure 5E). Transfection of an miR-206 mimic or inhibitor was used to, respectively, increase or decrease miR-206 expression levels in cells (Appendix A). Furthermore, RMRP-WT and RMRP-MUT were cloned into a luciferase reporter vector. Dual-luciferase reporter assays confirmed the direct binding of miR-206 to RMRP (Figure 5F). Moreover, mutation of the miR-206 binding sites in RMRP negatively affected the interaction between RMRP and miR-206 (Figure 5F).

We next observed that G6PD expression levels increased in BLCA cells following miR-206 inhibitor transfection, while they decreased with the addition of the miR-206 mimic (Figure 5G–I). We then predicted the binding sites of miR-206 and G6PD (Figure 5J). Additionally, dual-luciferase reporter assay confirmed the direct binding of miR-206 to G6PD, as mutation of the miR-206 binding sites in G6PD affected their interaction (Figure 5K). Notably, the observed decrease in G6PD expression levels following RMRP knockdown in BLCA cells was rescued when they were treated with the miR-206 inhibitor (Figure 5L,M). These data suggested that RMRP serves as an miR-206 sponge to regulate G6PD expression.

### 3.6. RMRP-Mediated Regulation of Tumor Progression in BLCA through the miR-206/G6PD axis

Next, we explored the effects of miR-206 on BLCA cell proliferation, migration, and invasion. CCK-8 and colony formation assays that inhibit miR-206 significantly promoted cell proliferation, whereas overexpressing miR-206 had the opposite effect (Figure 6A,B and Appendix A). Transwell assays revealed that the migratory and invasive capabilities of BLCA cells were improved with miR-206 inhibition but impaired with miR-206 overexpression (Figure 6C and Appendix A). Similarly, EdU levels were enhanced by miR-206 inhibition, but decreased with miR-206 overexpression (Appendix A). These data demonstrated the role of miR-206 in suppressing BLCA cell proliferation and invasive ability.

Then, we verified if RMRP can exert its tumor-promoting function through the miR-206/G6PD axis. The observed restricted BLCA cell proliferation caused by sh-RMRP was reversed with the addition of miR-206 inhibitor, as seen with CCK-8 and colony formation assays (Figure 6D,E). Similarly, the decreased EdU levels with RMRP knockdown were increased by inhibition of miR-206 (Figure 6F,G). The impaired migratory and invasive capabilities caused by sh-RMRP were also improved following miR-206 inhibition (Figure 6H,I). These data indicated that RMRP potentially regulates BLCA cell proliferation, migration and invasion via the miR-206/G6PD axis.

## 4. Discussion

BLCA is a common malignant urological tumor worldwide, and identifying new biomarkers to improve non-invasive detection and monitoring of BLCA is crucial. In this study, we examined the upregulated lncRNAs in urinary exosomes, leading to our proposed combination of RMRP in both plasma and urine exosomes as a prognostic biomarker for BLCA. Furthermore, we demonstrated that RMRP contributes to BLCA tumor progression via a novel ceRNA network: the miR-206/G6PD axis. Generally, miR-206 could target G6PD mRNA and restrict its transcription (Figure 6J). RMRP serves as a sponge for miR-206. Interactions between RMRP and miR-206 can impede miR-206 -mediated targeting of G6PD, which results in G6PD upregulation and thus increased the proliferation, invasive, and metastatic potentials of BLCA cells (Figure 6J).

Exosomal lncRNAs are promising diagnostic biomarkers for BLCA. In this study, we observed increased RMRP expression levels in urinary and plasma exosomes from BLCA patients compared with those from healthy individuals or benign urinary lesions by RNA-seq analysis. This high expression pattern was significantly associated with the clinicopathological characteristics of tumor stage (T), a poorer prognosis, and tumor grade in BLCA patients. We further explored the potential of exosomal RMRP as a diagnostic biomarker for BLCA. ROC analysis indicated that RMRP in plasma and urine exosomes was a promising diagnostic indicator with better diagnostic performance than the currently available clinical indicators (NMP-22, BTA, and urine cytology). In particular, the combined diagnosis of RMRP in urine and plasma exosomes had higher AUC (0.889) and sensitivity values, thereby improving diagnostic performance. In addition, we found that RMRP was stably present in exosomes from plasma or urine, likely from the protection of the membrane structure against external RNA enzymes and other substances that could cause RMRP degradation. Because of the stability offered by exosomes to RMRP and the ease of detecting RMRP in plasma and urine samples, we postulate that exosome-derived RMRP can serve as an early clinical biomarker for BLCA. However, there are certain limitations in clinical studies. For example, the limited number of clinical samples is insufficient to analyze BLCA patient prognosis. In the future, we will follow up with BLCA patients to further validate the diagnostic efficacy and complementary prognostic value of RMRP.

Recently, lncRNAs have been found to be related to tumor progression, invasion, and metastasis. The earliest known tumor related lncRNAs, such as H19, were found to enhance cell proliferation, inhibit tumor cell apoptosis, and promote tumor related angiogenesis and hypoxia tolerance [46]. As reported previously, RMRP is an oncogene and is widely expressed in different tissues of humans and mice [31]. Silencing RMRP can inhibit the proliferation of cholangiocarcinoma cells, stimulate apoptosis of these cells, and block G0/G1 cell cycles [47]. In our study, we confirmed that aberrant RMRP expression patterns are related to tumor progression and are involved in cell migration and invasion processes both in vitro and in vivo. RMRP often acts as a miRNA sponge to regulate other oncogenes. We also found G6PD expression levels to be upregulated in BLCA tissues compared with those in adjacent normal tissues. G6PD is involved in regulating cell transformation, proliferation, apoptosis, and angiogenesis [38,48]. Here, RMRP knockdown could downregulate G6PD expression. Therefore, G6PD is a possible target gene for RMRP to promote BLCA tumor formation. Meanwhile, based on previous reports [49,50], lncRNAs play a pivotal role in the regulation of gene expression and influence disease progression. HOTAIR, which is upregulated in various cancers and liver fibrosis, is a potential therapeutic target. Researchers have developed a dominant-negative lncRNA, HOTAIR-sbid, which disrupts harmful interactions and restores normal cell behavior, offering a promising therapeutic avenue with minimal side effects. Therefore, an RNA-based approach to counteract the function of RMRP in BLCA could be considered for clinical therapeutic purposes.

It is well-known that lncRNAs can act as ceRNAs by sponging miRNAs, which results in the regulation of downstream target gene expression. We explored the RMRP ceRNA network using miRDB and studied the regulatory role of RMRP on the miR-206/G6PD axis. We found that RMRP could negatively modulate miR-206 expression by direct interaction. Additionally, following RMRP silencing, the observed decreased G6PD expression levels could be rescued with miR-206 inhibition. Recently, miR-206 was reported to target the G6PD gene to regulate muscle cell proliferation [51,52]. These results suggested that RMRP can regulate G6PD expression via directly binding and sponging miR-206. Furthermore, downregulation of miR-206 with a specific inhibitor could significantly promote BLCA cell growth and migration. However, inhibition of miR-206 could improve the effects of RMRP knockdown on BLCA cell proliferation and migration. Overall, our study showed that RMRP is highly expressed in BLCA cells and acts as a sponge for miR-206 in the cytoplasm. RMRP-mediated sponging of miR-206 blocks the interaction between miR-206 and its target mRNA G6PD. This results in G6PD upregulation, which enhances BLCA cell migration and invasion. To date, we have not examined the effects of exosome-derived RMRP from BLCA cells or patients on BLCA development. This requires future exploration.

## 5. Conclusions

Our research first verified the high diagnostic efficacy of plasma and urinary exosome-derived RMRP in BLCA, then showed the stronger efficacy of the two combined. Furthermore, our data demonstrated that RMRP plays an oncogenic role in BLCA development via the miR-206/G6PD axis, which helped elucidate the mechanism of RMRP in BLCA progression. RMRP is therefore a promising biomarker for BLCA. Our study suggests that targeting the RMRP/miR-206/G6PD axis is a potential strategy for BLCA treatment.

## Figures and Tables

**Figure 1 cancers-15-05305-f001:**
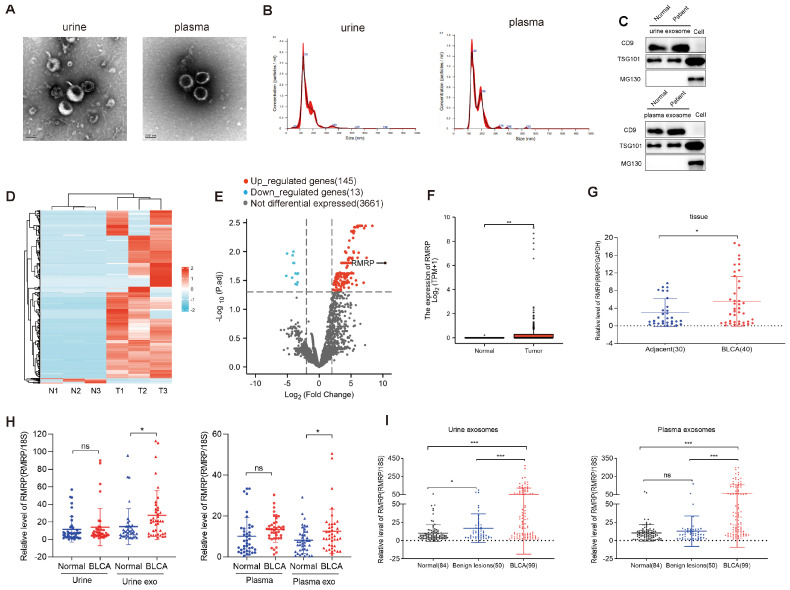
Urinary exosome RMRP was highly expressed in bladder cancer (BLCA) patients and associated with a poor prognosis. (**A**) Transmission electron microscopy (TEM) images showing the external features of exosomes isolated from urine and plasma samples; scale bar: 100 nm. (**B**) Nanoparticle tracking analysis (NTA) was performed to determine the size distribution of exosomes from urine and plasma samples. (**C**) Western blot analysis of CD9, TSG101, MG130 protein expression in urine exosomes, plasma exosomes and cells. The uncropped blots are shown in Appendix A. (**D**) A cluster heat map displaying the significantly dysregulated long non-coding RNAs (lncRNAs) from the urinary exosomes of healthy controls and BLCA patients. The red and blue strips represent high and low expression levels, respectively. (**E**) A volcano plot of the RNA-sequencing results. Blue dots represent downregulated genes and red dots represent upregulated genes in BLCA patients relative to healthy controls. N1-N3: healthy controls; T1-T3: BLCA patients; *p*.adj: adjusted *p*-value. (**F**) RMRP expression in BLCA and normal tissues in The Cancer Genome Atlas (TCGA) database. (**G**) RT-qPCR was performed to examine the relative RMRP expression levels in BLCA and adjacent sample tissues. RMRP expression was normalized to GAPDH expression. (**H**) RMRP expression in plasma, urine, urine and plasma exosomes from normal and BLCA patients. Urine exo: urine exosome, Plasma exo: plasma exosome. (**I**) RMRP expression was examined in exosomes extracted from urine and plasma samples from three groups of subjects. 18S RNA was used as an internal reference gene in exosomes. All data are presented as the mean ± standard error of the mean (SEM) of triplicate experiments. ns: not significant. *p* > 0.05, * *p* < 0.05, ** *p* < 0.01, *** *p* < 0.001.

**Figure 2 cancers-15-05305-f002:**
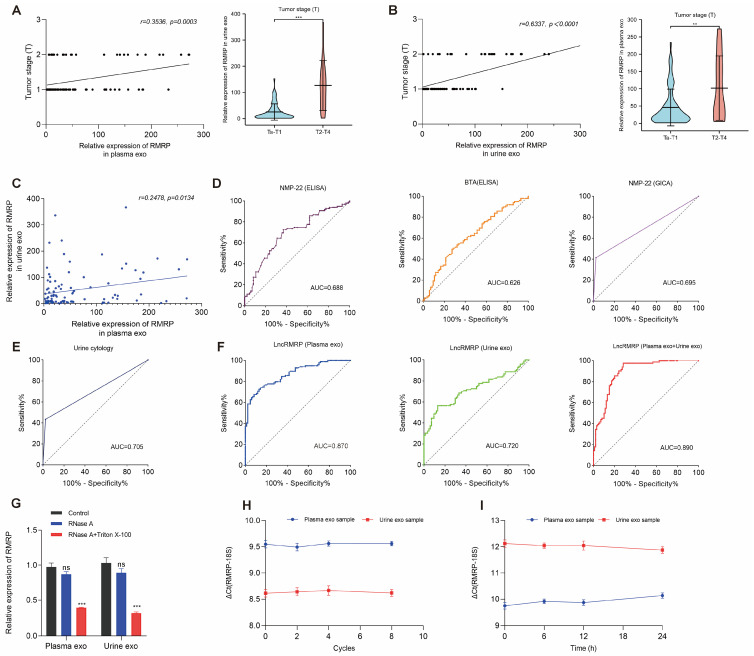
Diagnostic efficacy of RMRP in urinary and plasma exosomes in BLCA patients. (**A**,**B**) Correlation between RMRP expression levels in (**A**) plasma and (**B**) urine exosomes and tumor stage (T) of BLCA patients**.** (**C**) Correlation between RMRP expression levels in urine or plasma exosomes of BLCA patients (*p* = 0.0134). (**D**,**E**) Diagnostic performance of (**D**) NMP-22 (ELISA), BTA (ELISA), NMP-22 (GICA), and (**E**) urine cytology for BLCA patients. AUC: area under the curve. NMP-22: nuclear matrix protein 22. BTA: bladder tumor antigen; GICA: colloidal gold immunochromatography. (**F**) Receiver operating characteristic (ROC) curves were established by using plasma exosomal RMRP, urine exosomal RMRP, and the combination of RMRP in urine and plasma exosomes to explore their diagnostic potentials for BLCA patients. (**G**) RT-qPCR was used to determine RMRP expression levels in the plasma and urine exosome after treatment with 1μg/mL RNase A alone or combined with 0.1% Triton ×100 for 30 min. (**H**,**I**) The stability of RMRP expression in plasma and urine exosome was tested following (**H**) multiple freeze–thaw cycles and (**I**) prolonged incubation at room temperature. exo: exosome. All data are presented as the mean ± SEM of triplicate experiments. ns: no significance, ** *p* < 0.01, *** *p* < 0.001.

**Figure 3 cancers-15-05305-f003:**
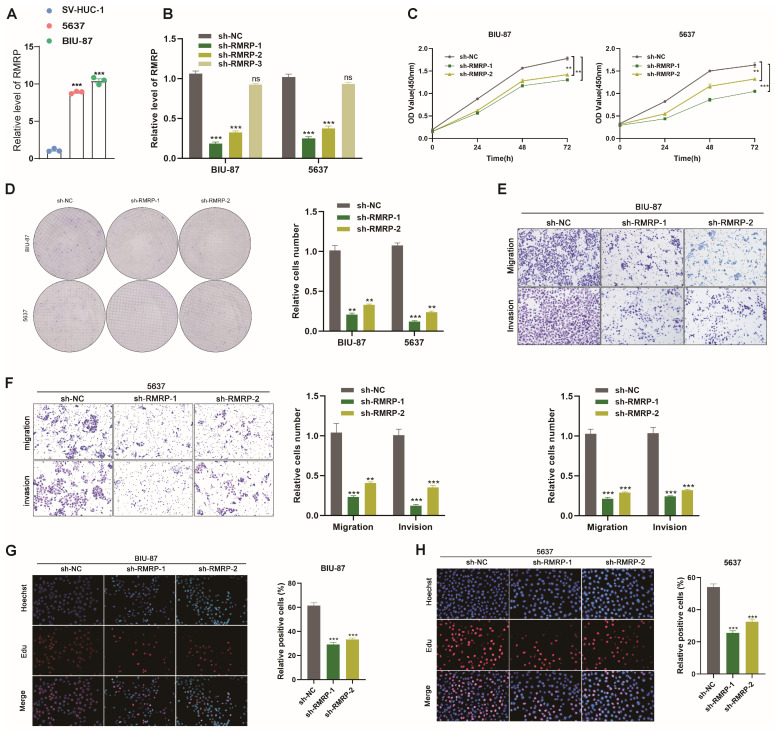
RMRP promotes proliferation, migration, and invasion in BLCA cells. (**A**) The relative expression of RMRP in 5637 and BIU-87 BLCA cells and control SV-HUC-1 cells. RMRP expression was normalized to GAPDH expression. (**B**–**H**) 5637 and BIU-87 cells were transfected with sh-NC or sh-RMRP. (**B**) qRT-PCR analyses of RMRP expression in 5637 and BIU-87 cells. (**C**) Cell proliferation was determined at 0, 24, 48 and 72 h after treatment. (**D**) Colony formation assays of 5637 and BIU-87 cells transfected with sh-NC or sh-RMRP for 14 days. (**E**,**F**) Cell migration and invasion assays using Transwell in 5637 and BIU-87 cells. (**G**,**H**) EdU assays were used to detect the proliferation rate of 5637 and BIU-87 cells after RMRP knockdown. Columns are the average of three independent experiments. All data are presented as the mean ± SEM of triplicate experiments. ns: no significance, ** *p* < 0.01, *** *p* < 0.001.

**Figure 4 cancers-15-05305-f004:**
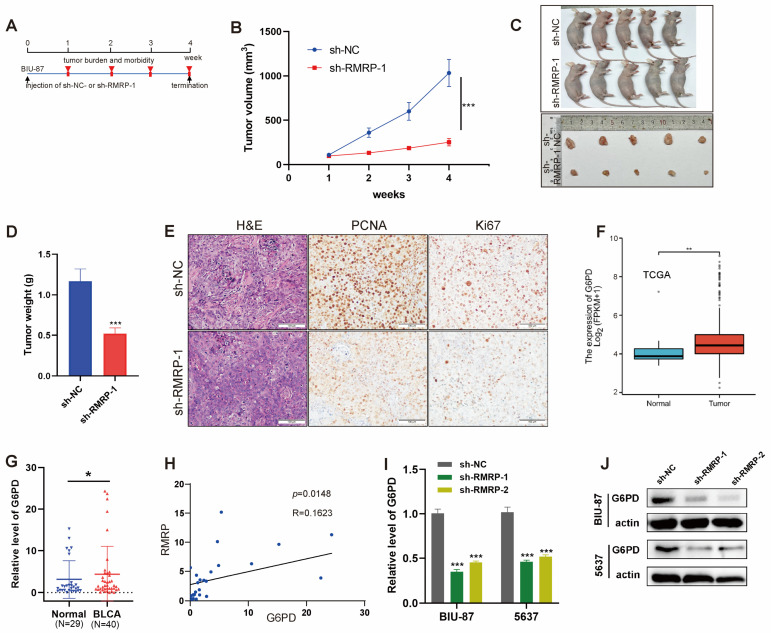
Positive correlation between RMRP and G6PD expression in BLCA Cells. (**A**–**E**) In vivo experimental scheme of BIU-87 allografts. (**A**) Experimental scheme of in vivo antitumor experiment (*n* = 5 mice per group). (**B**) The tumor volume of each treatment group. (**C**) Photos of mice and tumors from all groups. (**D**) The tumor weight at the end point of the animal experiment. (**E**) Representative images of H and E, PCNA, and Ki67 staining in the tumor, scale bar = 100 µm. (**F**) G6PD expression in BLCA and normal tissues in TCGA database. (**G**) G6PD expression in tumor tissues from normal and BLCA patients. (**H**) Correlation analysis of RMRP and G6PD expression in BLCA patients. Each dot represents one tumor tissue. (**I**) RT-qPCR and (**J**) WB analyses of G6PD expression in 5637 and BIU-87 cells after treatment of sh-NC or sh-RMRP. The uncropped blots are shown in Appendix A. All data are presented as the mean ± SEM of triplicate experiments. * *p* < 0.05, ** *p* < 0.01, *** *p* < 0.001.

**Figure 5 cancers-15-05305-f005:**
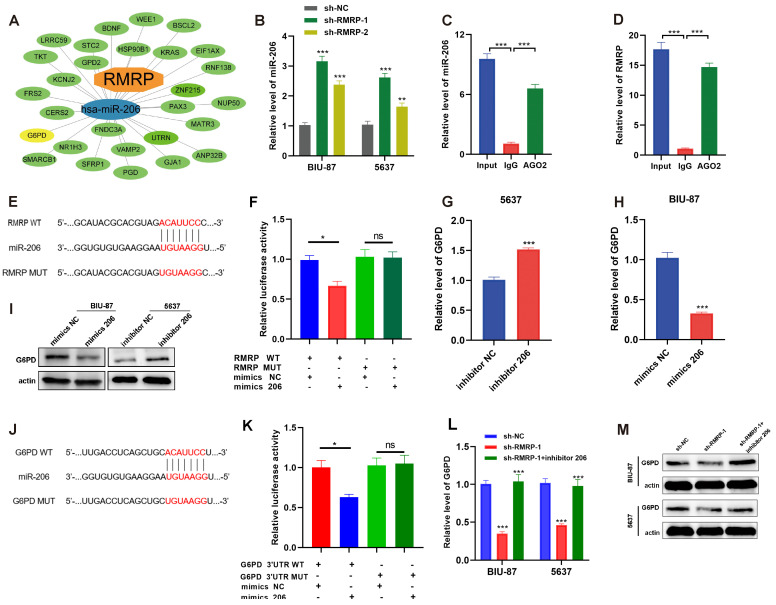
RMRP serves as an miRNA sponge of miR-206 to target G6PD. (**A**) The construction of a ceRNA network including RMRP, miR-206, and G6PD. (**B**) qRT-PCR analyses of miR-206 expression in 5637 and BIU-87 cells after treatment of sh-NC or sh-RMRP. (**C**,**D**) Anti-AGO2 RIP was performed followed by RT-PCR to detect the expression of (**C**) miR-206 or (**D**) RMRP associated with AGO2. (**E**) Schematic representation of the 3′-UTR of RMRP with the predicted target site for miR-206. The mutant site of RMRP 3′-UTR is indicated (without line). (**F**) Luciferase reporter constructs containing either RMRP WT or RMRP MUT at the predicted miR-206 target sequences were co-transfected into HEK293T cells, along with miR-206 or miR-NC mimics. (**G**) mRNA expression of G6PD in 5637 cells treated with inhibitor NC or inhibitor 206. (**H**) mRNA expression of G6PD in BIU-87 cells treated with mimics NC or mimics 206. (**I**) WB results showed the expression of G6PD in BLCA cells treated with mimic NC or mimic 206, and inhibitor NC or inhibitor 206. Actin was served as a loading control. (**J**) Schematic representation of the 3′-UTR of G6PD with the predicted target site for miR-206. The mutant site of G6PD 3′-UTR is indicated (without line). (**K**) Luciferase reporter constructs containing either G6PD WT or G6PD MUT at the predicted miR-206 target sequences were co-transfected into HEK293T cells, along with miR-206 or miR-NC mimics. (**L**) mRNA and (**M**) protein expression of G6PD in BIU-87 and 5637 cells treated with sh-NC, sh-RMRP or sh-RMRP + inhibitor 206. The uncropped blots are shown in Appendix A. All data are presented as the mean ± SEM of triplicate experiments. ns: no significance, * *p* < 0.05, ** *p* < 0.01, *** *p* < 0.001.

**Figure 6 cancers-15-05305-f006:**
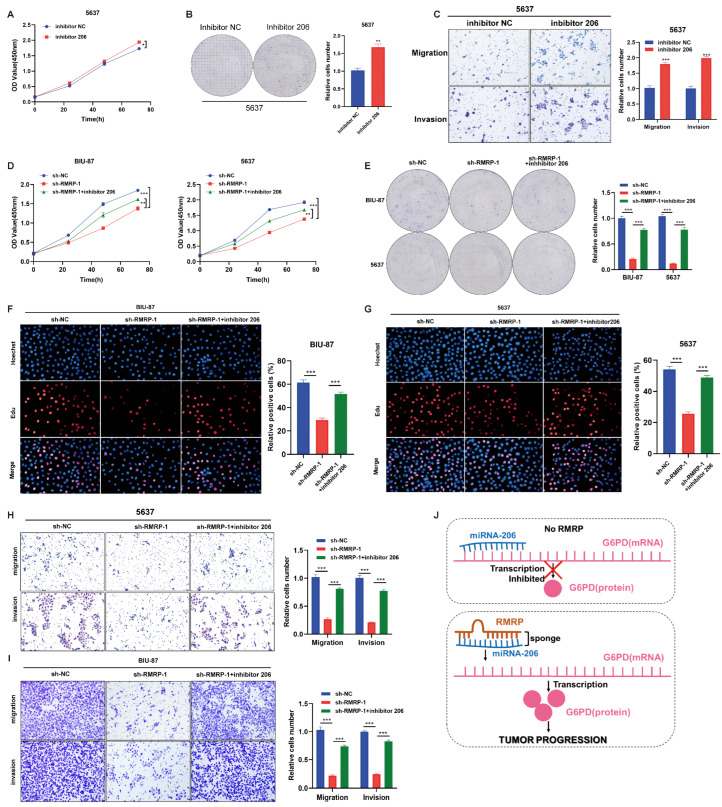
Regulation of RMRP on tumor progression in BLCA through miR-206/G6PD axis. (**A**) Cell proliferation of 5673 cells was determined at 0, 20, 40, 60 and 80 h after inhibitor NC or inhibitor 206 treatment. (**B**) Colony formation assays of 5637 cells treated with inhibitor NC or inhibitor 206 for 14 days. (**C**) Cell migration and invasion assays using Transwell in 5637 cells treated with inhibitor NC or inhibitor 206. (**D**–**I**) BIU-87 and 5673 cells were treated with sh-NC or sh-RMRP. Then, BIU-87 and 5673 cells with RMRP knockdown had the appearance of inhibitor 206. (**D**) Cell proliferation of BIU-87 and 5673 cells was determined by CCK-8. (**E**) Colony formation assays of BIU-87 and 5673 cells were detected at day 14 in different treatment groups. (**F**,**G**) EdU assays were used to detect the proliferation rate of BIU-87 and 5673 cells in different treatment groups. Columns are the average of three independent experiments. (**H**,**I**) Cell migration and invasion assays using Transwell in BIU-87 and 5673 cells. (**J**) Schematic showing the functional and molecular mechanisms of RMRP/miR-206/G6PD in tumor progression of BLCA. All data are presented as the mean ± SEM of triplicate experiments. * *p* < 0.05, ** *p* < 0.01, *** *p* < 0.001.

**Table 1 cancers-15-05305-t001:** Correlation between RMRP expression levels in urinary or plasma exosomes and clinicopathological characteristics of BLCA patients.

Characteristics	Case	Expression of Plasma Exosomal RMRP		χ^2^	*p* Value	Expression of Urine Exosomal RMRP		χ^2^	*p* Value
		High	Low			High	low		
All cases	99	50	49			50	49		
Gender									
Male	72	33 (45.83%)	39 (54.17%)	2.305	0.129	33 (45.83%)	39 (54.17%)	2.305	0.129
Female	27	17 (62.96%)	10 (37.04%)			17 (62.96%)	10 (37.04%)		
Age (years)									
≤65	47	22 (46.81%)	25 (53.19%)	0.489	0.484	22 (46.81%)	25 (53.19%)	0.489	0.484
>65	52	28 (53.45%)	24 (46.15%)			28 (53.45%)	24 (46.15%)		
Tumor size (cm)									
≤3	68	37 (54.41%)	31 (45.59%)	1.326	0.250	30 (44.11%)	38 (55.89%)	3.544	0.600
>3	31	13 (41.94%)	18 (58.06%)			20 (64.52%)	11 (35.48%)		
Tumor number									
≤1	80	39 (48.75%)	41 (51.25%)	0.514	0.323	41 (51.25%)	39 (48.75%)	0.093	0.803
>1	19	11 (57.89%)	8 (42.11%)			9 (46.37%)	10 (52.63%)		
Tumor stage (T)									
Ta–T1	73	32 (43.84%)	41 (56.16%)	4.946	0.026	27 (36.99%)	46 (63.01%)	20.322	<0.001
T2–T4	26	18 (69.23%)	8 (30.77%)			23 (88.46%)	3 (11.54%)		
Lymph node metastasis									
Negative	88	42 (46.66%)	46 (53.33%)	2.445	0.118	44 (50.00%)	44 (50.00%)	0.810	0.514
Positive	11	8 (60%)	3 (40%)			6 (54.55%)	5 (45.45%)		
Tumor grade									
Low	16	12 (71.43%)	4 (28.57%)	4.581	0.032	8 (50.00%)	8 (50.00%)	0.002	0.590
Middle-High	83	38 (47.06%)	45 (52.94%)			42 (50.60%)	41 (49.40%)		

**Table 2 cancers-15-05305-t002:** Diagnostic efficacy of NMP-22 (ELISA), BTA (ELISA), NMP-22 (GICA), urine cytology, RMRP in urinary exosomes, and RMRP in plasma exosomes.

Indicators	Youden Index	Cut-Off	AUC	AUC 95% CI	Sensitivity (%)	Specificity (%)	PPV (%)	NPV (%)
NMP-22 (ELISA)	0.358	1.739	0.688	0.611–0.765	63.1	72.7	69.9	66.2
BTA (ELISA)	0.221	7.140	0.626	0.544–0.707	72.6	49.5	68.1	55.0
NMP-22 (GICA)	0.390	1.500	0.695	0.644–0.747	41.4	97.6	58.6	95.3
Urine cytology	0.411	1.500	0.705	0.654–0.757	43.4	97.6	59.4	95.6
RMRP (Plasma exo)	0.596	11.774	0.870	0.820–0.919	86.9	72.7	86.7	73.0
RMRP (Urine exo)	0.435	16.168	0.720	0.646–0.794	86.9	56.6	83.6	62.9
RMRP (Plasma exo + Urine exo)	0.693	0.427	0.889	0.842–0.936	97.6	71.7	97.3	74.5

NMP-22: nuclear matrix protein 22; BTA: bladder tumor antigen; GICA: colloidal gold immunochromatography; AUC: area under the curve; CI: confidence interval; PPV: positive predictive value; NPV: negative predictive value.

## Data Availability

The datasets supporting the conclusions of this article are included within the article.

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
