# Peer review of "Exosomal Long Non-Coding Ribonucleic Acid Ribonuclease Component of Mitochondrial Ribonucleic Acid Processing Endoribonuclease Is Defined as a Potential Non-Invasive Diagnostic Biomarker for Bladder Cancer and Facilitates Tumorigenesis via the miR-206/G6PD Axis"

_cancers, 2023, doi:10.3390/cancers15215305_

Round 1

Reviewer 1 Report

Comments and Suggestions for Authors

The authors analyzed and compared the expression of exosomal lncRNA RMRP in the exosomes of urine exosomes and plasma exosomes from bladder cancer patients.

Additionally, authors showed that RMRP is associated with tumor progression and participates in migration and invasion processes through the miR-206/G6PD axis both in vitro and in vivo.

I think most of the process is well explained.

However, I think the authors should add the following proof.

1. It must be proven whether the effect of RMRP and miR206 regulation by siRNA, mimic, or inhibitor is maintained in exosomes. Additionally, it must be shown whether they are involved in phenotype and gene regulation in BIU-87 and 5637 cells.

2. In urine normal samples, the expression of RMBP ranges from a maximum of 30.86 to a minimum of 20.49. This is a huge difference. In other words, the variation between samples is large. Authors must confirm the status of exsomal RNA used in their experiments.

Additionally, such a large difference in bladder cancer diagnosis does not mean that RMBP can be considered a diagnostic marker. Authors should strongly describe these limitations.

Comments on the Quality of English Language

Well written.

Author Response

Thank you very much for taking the time to review this manuscript. Here are our responses.

Comment 1: It must be proven whether the effect of RMRP and miR206 regulation by siRNA, mimic, or inhibitor is maintained in exosomes. Additionally, it must be shown whether they are involved in phenotype and gene regulation in BIU-87 and 5637 cells.

Response 1: Thank you very much for your valuable comments. Our study demonstrated the diagnostic value of RMRP in blood and urine exosomes for bladder cancer. Previously, we have definite the effect of RMRP and miR206 in regulating phenotype and gene expression in BIU-87 and 5637 cells, not in exosomes. Partly due to the long period of extraction and identification of exosomes, we will explore the effect of RMRP and miR206 in exosomes in the future. The results will be presented in our subsequent work.

Comment 2:  In urine normal samples, the expression of RMBP ranges from a maximum of 30.86 to a minimum of 20.49. This is a huge difference. In other words, the variation between samples is large. Authors must confirm the status of exosomal RNA used in their experiments. Additionally, such a large difference in bladder cancer diagnosis does not mean that RMBP can be considered a diagnostic marker. Authors should strongly describe these limitations.

Response 2: Thank you for bringing up this question. This difference in RMRP expression in clinical samples might be due to individual variations and sample collection. The urine samples we collected were random urine, with varying concentrations and densities. As a result, we observed differences in the CT values of RMRP expression, both in normal healthy and BLCA patient urine samples. To address this, 18S was as a control gene to normalize the data. Similar approaches using 18S as a control gene have been reported in previous studies (PMID: 35127956).

Reviewer 2 Report

Comments and Suggestions for Authors

In this paper, the authors focus on the role of EV and lncRNA RMRP in BLCA.  They find that RMRP is expressed in exosomes from tissue, urine and plasma from patients with respect to healthy control and that it is associated with the clinicopathological characteristics of tumor stage, poorer prognosis and tumor grade. Mechanistically, this lncRNA relates to tumor progression and is involved in cell migration/ invasion via miR-206/G6PD axis.

Overall, the authors propose RMRP as a useful tool in the non-invasive diagnosis of BLCA, representing a good candidate for the therapy.

The paper is interesting; however, in the present form it suffers of some criticisms that should be addressed.

Major points:

1)    Some general sentences regarding the guidelines for the preparation of the manuscript should be deleted.

2)    In the characterization of exosomes by western blot, the authors should use the same markers and should add a negative control for exosomes (e.g. calnexin) and a tissue sample should be shown.

3)    Figure 3 images have a low resolution that renders the reading of the data difficult. Please replace the figures.

4)    The authors write that: transfection with sh-RMRP markedly reduced the expression of miR-206 (Fig. 5B). This is not in line with the proposed ceRNA role of RMRP and is opposite to the shown data.

5)    Luciferase experiments show a weak decrease of luciferase activity in response to miR-206 mimics. This raises doubts about the robustness of the data. Moreover, the luc activity for the mutated vector is lower than the wild-type. These data should be clarified and experimentally confirmed.

6)    Quantifications of western blot signals should be reported, and less exposed images should be shown in order to better appreciate the differences among the experimental samples.

7)     The authors show a significant enrichment of RMRP in exosomes from BLCA patients with respect to normal subjects. This is the most interesting aspect of the manuscript, in my opinion, in a diagnostic perspective. The authors should perform functional assays in cells receiving these vesicles in order to investigate their role on target cells proliferation, invasion, migration, cloning abilities.

8)    In the light of the proposed role of the lncRNA RMRP as ceRNA and potentially as interactor of proteins, the authors should discuss an RNA-based approach to counteract its function (for refs see also Zhu et al., 2022; Garbo et al., 2022).

Minor points:

Typing errors should be amended.

Reviewer 3 Report

Comments and Suggestions for Authors

This study provides information of scientific value and deserves to be published. Only some minor changes are recommended to correct some small errors in the writing or to make its reading more attractive to researchers interested in the topic.

Minor corrections (but a lot!)

1.- Please rewrite the abstract. There are sentences that do not correspond to the work

Lines 22-24  We analyzed and compared the different RMRP expression in exosomes from tissue samples, urine exosomes, and plasma exosomes from three BLCA patients and three healthy ? using RNA sequencing.

REMARK: The RNAseq is a general technique from which, after differential expression analysis, you select RMRP as the most upregulated lncRNA

Lines 25-27. “Then, we explored the mechanism of RMRP on the development of bladder cancer in vivo and  in vitro. In this study, we observed increased RMRP expression in urinary- and plasma-exosomes of BLCA patients compared with healthy individuals.“

REMARK: the sentence underlined should be before the other, and it should be convenient to say how (which techniques) the in vitro and in vivo  mechanisms are tested.

Lines 27-29 : “High RMRP expression was RMRP was stably expressed in plasma- and urine-exosomes and significantly associated with the clinicopathological  characteristics of tumor stage, poorer prognosis and tumor grade in BLCA patients”

REMARK: The initial four words make no sense in the sentence. The stability in plasma and urine has not been proved in the work. What has been proved is the stability of the lncRNA in the isolated exosomes.

Lines 31-32: “Furthermore, RMRP related to tumor progression and involved in the process of mi-31 gration and invasion via miR-206/G6PD axis both in vitro and in vivo”

REMARK: ““Furthermore, RMRP is related…

 2.- The whole text

Along all the text. Please introduce “space” before de numerical reference (¡all 52!)

ie. Line 49, “BLCA has become 49 one of the ten most common cancers worldwide[1]”

should be: “BLCA has become 49 one of the ten most common cancers worldwide [1]”

3.- Introduction

.- lines 40-47: 

REMARK: Please delete the description of the section

.- lines 68-70  Additionally, exosomal lncRNAs are stable in blood and fluid to have the capacity to distinguish individuals with tumors or healthy[14]

REMARK: Please re-write this sentence, it has no sense

.-Lines 98-100 “whether RMRP can be used as a potential biomarker for the diagnosis of BLCA, and to study…”

REMARK: Not before discover its deregulation!

4.- Materials and methods

.-Lines 120-122: “The results showed the presence of CD9 and TSG101 expression in urine exosomes and CD54 and TSG101 expression in plasma exosomes (Fig. 1C).

REMARK: Please, explain the significance of CD9 and CD54 on each

.- Line 125: “Solution A and Solution B”

REMARK: Please, indicate composition or reference to the kit

.- Line 243  “The RIP assay 243 utilized antibodies such as anti-AGO2 and control IgG (Millipore, USA).”

REMARK: Please explain the significance of anti-AGO2 in the RIP design

.- Line 272 “Tumor volumes = length×width2/2”

REMARK: Please, correct the formula

.- Line 278; H2O2

REMARK: Please, correct the formula

5.- RESULTS

.-Lines 314-319: “Of note, there was no significant difference of RMRP expression in blood and urine samples (Fig.1H). However, RMRP expression in exosomes of urinary and plasma was  increased in BLCA patients than the normal. (Fig.1H). Additionally, expanding sample  amounts, exosomes-RMRP expressions from urinary and plasma were significantly higher in BLCA group compared with the normal and benign lesions groups (Fig. 1I, supplementary Table S4).”

REMARK: Please, correct this section for clarity. Some sentences are ambiguous

.- Line 354: The Table1  needs peer editing

.- Line 370: “As better stability is a critical prerequisite for tumor markers[37],”

REMARK: No comparative is done. Please change “better” for high or good

.- Line 394 “The tumor volume of sh-RMRP group grew more slowly than those in sh-NC group (Fig.4B), consist with the weight of residual…”

REMARK; the link of sentences is wrong. Please Change “consist” to “accordingly or in accordance with”

Comments on the Quality of English Language

Revise carefully.

Author Response

Thank you very much for taking the time to review this manuscript. Here are our responses.

Comment 1: Please rewrite the abstract. There are sentences that do not correspond to the work.

Lines 22-24 We analyzed and compared the different RMRP expression in exosomes from tissue samples, urine exosomes, and plasma exosomes from three BLCA patients and three healthy. using RNA sequencing.

REMARK: The RNAseq is a general technique from which, after differential expression analysis, you select RMRP as the most upregulated lncRNA

Lines 25-27. “Then, we explored the mechanism of RMRP on the development of bladder cancer in vivo and in vitro. In this study, we observed increased RMRP expression in urinary- and plasma-exosomes of BLCA patients compared with healthy individuals.”

REMARK: the sentence underlined should be before the other, and it should be convenient to say how (which techniques) the in vitro and in vivo mechanisms are tested.

Lines 27-29: “High RMRP expression was RMRP was stably expressed in plasma- and urine-exosomes and significantly associated with the clinicopathological characteristics of tumor stage, poorer prognosis and tumor grade in BLCA patients”

REMARK: The initial four words make no sense in the sentence. The stability in plasma and urine has not been proved in the work. What has been proved is the stability of the lncRNA in the isolated exosomes.

Lines 31-32: “Furthermore, RMRP related to tumor progression and involved in the process of mi-31 gration and invasion via miR-206/G6PD axis both in vitro and in vivo”

REMARK: ““Furthermore, RMRP is related…

Response 1: Thank you for your suggestion. We have revised the abstract as follows:

Bladder cancer (BLCA) is one of the highly sensitive and specific non-invasive tumor biomarkers to facilitate early diagnosis. Exosome-derived long non-coding RNAs (lncRNAs) hold promise as diagnostic biomarkers for BLCA. In this study, we employed RNA sequencing to compare the expression patterns of lncRNAs in urine exosomes from three BLCA patients and three healthy individuals. RMRP displayed the most significant differential expression. Elevated RMRP expression levels were observed in urinary and plasma exosomes from BLCA patients compared with those from healthy individuals. RMRP exhibited significant associations with certain BLCA patient clinicopathological features, including tumor stage, poor prognosis, and tumor grade. Combined diagnosis using RMRP in urine and plasma exosomes demonstrated superior diagnostic performance with receiver operating characteristic curve analysis. RMRP was found to be related to BLCA tumor progression and the cell migration and invasion processes via the miR-206/G6PD axis both in vitro and in vivo. Mechanistically, RMRP serves as a miR-206 sponge, as suggested by dual-luciferase reporter assays and RNA immunoprecipitation. Our study suggests that the combined diagnosis of RMRP in urinary and plasma exosomes can serve as an excellent non-invasive diagnostic biomarker for BLCA patients. Additionally, targeting the RMRP/miR-206/G6PD axis holds promise as a therapeutic strategy for BLCA.

Comment 2: The whole text

Along all the text. Please introduce “space” before de numerical reference (¡all 52!)

  1. Line 49, “BLCA has become 49 one of the ten most common cancers worldwide [1]”

should be: “BLCA has become 49 one of the ten most common cancers worldwide [1]”

Response 2: Thank you for your suggestion. We have made corrections to such issues in the revised manuscript.

Comment 3: Introduction

.- lines 40-47: 

REMARK: Please delete the description of the section

.- lines 68-70  Additionally, exosomal lncRNAs are stable in blood and fluid to have the capacity to distinguish individuals with tumors or healthy[14]

REMARK: Please re-write this sentence, it has no sense

.-Lines 98-100 “whether RMRP can be used as a potential biomarker for the diagnosis of BLCA, and to study…”

REMARK: Not before discover its deregulation!

Response 3: Thank you for your suggestion. The beginning of the manuscript's introduction has been revised as per your suggestions.

Line 41-51: “Bladder cancer (BLCA) is a highly invasive malignant tumor that affects the urinary system. Data from 2023 indicate that BLCA has become one of the top ten most prevalent cancers globally [1]. Currently, the clinical use of biomarkers, such as nuclear matrix protein 22 (NMP-22) and bladder tumor antigen (BTA), suffers from low sensitivity and the potential for false positives. While cystoscopy remains the gold standard for diagnosing BLCA, it is an invasive procedure. Additionally, BLCA is often asymptomatic in its early stages, making cystoscopy an unsuitable early screening tool [5,6]. Patients in the early stages of BLCA typically do not exhibit noticeable symptoms, emphasizing the critical importance of early BLCA diagnosis for good prognosis [7,8]. Therefore, the identification of promising early molecular biomarkers holds significant potential for a better understanding of BLCA pathogenesis and enhancing its clinical treatment.”

Comment 3: Materials and methods

.-Lines 120-122: “The results showed the presence of CD9 and TSG101 expression in urine exosomes and CD54 and TSG101 expression in plasma exosomes (Fig. 1C).

REMARK: Please, explain the significance of CD9 and CD54 on each

.- Line 125: “Solution A and Solution B”

REMARK: Please, indicate composition or reference to the kit

.- Line 243  “The RIP assay 243 utilized antibodies such as anti-AGO2 and control IgG (Millipore, USA).”

REMARK: Please explain the significance of anti-AGO2 in the RIP design

.- Line 272 “Tumor volumes = length×width2/2”

REMARK: Please, correct the formula

.- Line 278; H2O2

REMARK: Please, correct the formula

Response 4: Thank you for your suggestion. We have made corrections in the revised manuscript as mentioned above.

As follows:

- Line 117: “Next, the supernatant was mixed with Solution A and Solution B from the exosome ex-traction kit, then stored overnight at 4°C.”

- Line 240: “AGO2 is an essential component of the RNA-induced silencing complex (RISC).”

- Line 267: “Tumor volumes were calculated using the following formula: Tumor volume = (length×width2)/2.”

- Line 274: “Firstly, the sections were deparaffinized and rehydrated, followed by exposure to 3% H2O2 in methanol to block endogenous peroxidase activity.”

Comment 5: RESULTS

.-Lines 314-319: “Of note, there was no significant difference of RMRP expression in blood and urine samples (Fig.1H). However, RMRP expression in exosomes of urinary and plasma was increased in BLCA patients than the normal. (Fig.1H). Additionally, expanding sample amounts, exosomes-RMRP expressions from urinary and plasma were significantly higher in BLCA group compared with the normal and benign lesions groups (Fig. 1I, supplementary Table S4).”

REMARK: Please, correct this section for clarity. Some sentences are ambiguous

.- Line 354: The Table1  needs peer editing

.- Line 370: “As better stability is a critical prerequisite for tumor markers[37],”

REMARK: No comparative is done. Please change “better” for high or good

.- Line 394 “The tumor volume of sh-RMRP group grew more slowly than those in sh-NC group (Fig.4B), consist with the weight of residual…”

REMARK; the link of sentences is wrong. Please Change “consist” to “accordingly or in accordance with”

Response 5: Thank you for reviewing this manuscript so carefully and pointing out many issues that we had overlooked. We have made corrections or additions to the revised manuscript according to your comments.

For example:

Line 371: “Because good stability is a critical prerequisite for tumor biomarkers [35], we next verified the stability of RMRP in plasma and urinary exosomes.”

Line 397: “Tumors of the sh-RMRP group grew more slowly than those of the sh-NC group (Fig. 4B), accordingly the volumes and weights of the residual tumors (Fig. 4C, D).”

Round 2

Reviewer 2 Report

Comments and Suggestions for Authors

None